# Hilab System Device in an Oncological Hospital: A New Clinical Approach for Point of Care CBC Test, Supported by the Internet of Things and Machine Learning

**DOI:** 10.3390/diagnostics13101695

**Published:** 2023-05-11

**Authors:** Aléxia Thamara Gasparin, Claudiane Isabel Franco Araujo, Mônica Ribas Cardoso, Patricia Schmitt, Juliana Beker Godoy, Eduarda Silva Reichert, Maria Eduarda Pimenta, Caroline Bretas Gonçalves, Erika Bergamo Santiago, Ivan Lucas Reis Silva, Bruno de Paula Gaideski, Milena Andreuzo Cardoso, Fernanda D’Amico Silva, Viviane da Rosa Sommer, Luis Felipe Hartmann, Carolina Rodrigues de Araujo Perazzoli, João Samuel de Holanda Farias, Olair Carlos Beltrame, Nicole Winter, Diego Rinaldi Pavesi Nicollete, Silvia Nathalia Bueno Lopes, João Victor Predebon, Bernardo Montesanti Machado de Almeida, Sérgio Renato Rogal Júnior, Marcus Vinícius Mazega Figueredo

**Affiliations:** 1Department of Research and Development, Hilab, Jose Altair Possebom, 800, Curitiba 81270-185, PR, Brazil; claudiane.araujo@hilab.com.br (C.I.F.A.); monica.cardoso@hilab.com.br (M.R.C.); patricia.schmitt@hilab.com.br (P.S.); beker.juliana@gmail.com (J.B.G.); eduarda.reichert@hilab.com.br (E.S.R.); maria.pimenta@hilab.com.br (M.E.P.); caroline.goncalves@hilab.com.br (C.B.G.); erikasantiago23@gmail.com (E.B.S.); ivan.silva@hilab.com.br (I.L.R.S.); bgaideski@gmail.com (B.d.P.G.); milena.cardoso@hilab.com.br (M.A.C.); fernanda.silva@hilab.com.br (F.D.S.); viviane.sommer@hilab.com.br (V.d.R.S.); luis.hartmann@hilab.com.br (L.F.H.); carolina.perazzoli@hitechnologies.com.br (C.R.d.A.P.); diego.nicollete@hilab.com.br (D.R.P.N.); silvia.lopes@hilab.com.br (S.N.B.L.); joao.predebon@hilab.com.br (J.V.P.); bernardo@hilab.com.br (B.M.M.d.A.); marcus@hilab.com.br (M.V.M.F.); 2Erasto Gaertner Hospital, Curitiba 81520-060, PR, Brazil; jshfarias@gmail.com (J.S.d.H.F.); obeltrame@erastogaertner.com.br (O.C.B.); nwinter@erastogaertner.com.br (N.W.)

**Keywords:** mobile device, rbc count, wbc count, platelets count, fast cbc test, AI hematology, POC hematology

## Abstract

The complete blood count (CBC) is a highly requested test that is generally restricted to centralized laboratories, which are limited by high cost, being maintenance-demanding, and requiring costly equipment. The Hilab System (HS) is a small, handheld hematological platform that uses microscopy and chromatography techniques, combined with machine learning (ML) and artificial intelligence (AI), to perform a CBC test. This platform uses ML and AI techniques to add higher accuracy and reliability to the results besides allowing for faster reporting. For clinical and flagging capability evaluation of the handheld device, the study analyzed 550 blood samples of patients from a reference institution for oncological diseases. The clinical analysis encompassed the data comparison between the Hilab System and a conventional hematological analyzer (Sysmex XE-2100) for all CBC analytes. The flagging capability study compared the microscopic findings from the Hilab System and the standard blood smear evaluation method. The study also assessed the sample collection source (venous or capillary) influences. The Pearson correlation, Student *t*-test, Bland–Altman, and Passing–Bablok plot of analytes were calculated and are shown. Data from both methodologies were similar (*p* > 0.05; r ≥ 0.9 for most parameters) for all CBC analytes and flagging parameters. Venous and capillary samples did not differ statistically (*p* > 0.05). The study indicates that the Hilab System provides humanized blood collection associated with fast and accurate data, essential features for patient wellbeing and quick physician decision making.

## 1. Introduction

The complete blood count (CBC) is the test that provides physicians the most information about patient homeostasis and is also one of the most requested tests in clinical laboratories [1]. This test evaluates red blood cells (RBC), hemoglobin (HB), hematocrit (HT), mean corpuscular volume (MCV), mean corpuscular hemoglobin (MCH), mean corpuscular hemoglobin concentration (MCHC), and platelet (PLT), white blood cell (WBC), neutrophil (NEU), lymphocyte (LIN), monocyte (MON), eosinophil (EOS) and basophil (BAS) counts. Thus, physicians usually request CBC for diagnosing diseases, evolutionary control of infections and cancers, medical emergencies, surgeries, traumatology, acute hemorrhagic states, allergies, and monitoring for chemotherapy and radiotherapy [2]. Additionally, with the recent SARS-CoV-2 pandemic, the CBC test presented a higher impact as a complementary laboratory tool, especially in developing and third-world countries that financially struggled to test their populations [3,4].

Clinical laboratories employ large devices with flow cytometry or resistivity-impedance methodologies (hematological analyzers) to perform the CBC test. However, these types of equipment require a high initial cost, and a structured laboratory environment, besides being frequent and expensive quality control processes [5]. In addition, these analyzers do not confirm hematological alterations, requiring subsequent microscopic evaluation by a trained and qualified professional. These data justify the average turnaround time (TAT) of twenty-four hours to the conventional CBC test and demonstrate how point of care testing (POCT) is a relevant tool in hematology laboratory practice [6,7,8,9].

Regarding the POCT, machine learning (ML) is revolutionizing the healthcare industry by enabling more accurate and efficient diagnostics. ML algorithms can analyze large amounts of medical data, including images, patient records, and other sources of information, and identify patterns and correlations that may be difficult for human experts to detect. Thus, this technique can help doctors and researchers to identify new diseases, develop new treatments, and improve patient outcomes. In this sense, ML is being used to help diagnose and track the spread of COVID-19 [10,11], considered a critical need during this pandemic. ML is also used in other disease diagnostics, such as polycystic ovary syndrome (PCOS), and various blood and biochemical tests [11,12]. Thus, through patient data analysis, these algorithms can help to identify the early signs of a wide range of diseases, providing earlier clinical intervention.

The most widely used handheld devices in the CBC test field include the measurement of HB concentration. Recently, other devices have been introduced, quantifying RBC, LEU, and PLA, besides some RBC indices (MCV, MCH, MCHC). However, most of these devices present high costs and are not liable to calibration or control procedures, which results in poor quality compared to standard instruments [10,11,13]. The Hilab System is a novel point-of-care hematological platform (National Health Surveillance Agency (ANVISA) registration no. 80583710018 and 80583710007 (Figure 1) that uses microscopy and chromatography techniques, combined with ML and artificial intelligence (AI). These handheld devices are factory calibrated and accompany simple operation single-use diagnostic kits for blood collection and preparation, accepting both venous and fingerstick blood samples (containing or not K3EDTA).

The first clinical validation study of the Hilab System evaluated 450 blood samples of majorly healthy volunteers, invited to study participation in the Hi Technologies dependencies. The authors analyzed the samples in the Hilab System and a conventional hematological analyzer (Sysmex XE-2100; CBC reference values) and compared the obtained data. As a result, both methodologies presented high accuracy and correlation results for all the evaluated parameters. However, although satisfactory results were present in the first validation work, this study analyzed few samples with expressive hematological alterations, limiting the understanding of the device performance in patients with this profile. Therefore, the present study assessed the accuracy, flagging capabilities, and the sample source influence of this new POCT hematological solution, compared to the CBC results provided by Sysmex XE-2100 (Sysmex Corporation, Tokyo, Japan) in Erasto Gaertner Hospital, a reference institution for the diagnosis and treatment of oncological patients.

## 2. Materials and Methods

### 2.1. Experimental Design and Written Informed Consent

Between March and August of 2022, the study participants were approached and asked about their interest in research participation at the Erasto Gaertner Hospital premises (Curitiba, Parana, Brazil), a reference institution for the diagnosis and treatment of oncological patients. The study realization encompassed three hospital sectors, where the “Blood bank” represented the healthy group of patients (“healthy” individual; criteria described in Ordinance n°. 158, Brazil Ministry of Health, 4 February 2016) and the “Emergency room” the group of patients with clinical alterations or previous pathologies. The “hematology and BMT (bone marrow transplant) intensive care unit” was the last sector evaluated, as the hematological diseases group of patients. All subjects or their legal guardian(s) provided informed written consent for study participation.

### 2.2. The Hilab System Device

The Hilab Lens and Hilab Flow form the Hilab System (Figure 1). The first device carries out the optical microscopy technique, weighing only 500 g, with a footprint of 19.7 × 9.9 × 15.2 cm (National Health Surveillance Agency (ANVISA) registration number 80583710018). This device has a Snapdragon processor to run AI algorithms on the device, enabling offline operation, and presents two stages of sample analysis, in which the first read the WBC (in the first chamber of the disposable hemocytometer; see Section “Sample preparation process”) and the second the RBC and PLT (in the second chamber of the disposable hemocytometer; see the Section “Sample preparation process”). At each stage, the Hilab Lens uses autofocus and image capture processes to acquire over 400 images of each sample, which are attached to form the final figure of the cells. Hilab Flow performs the chromatography technique, weighing only 450 g, with a footprint of 12.4 × 12.4 × 12.7 cm (ANVISA registration number 80583710007) and a versatile point-of-care testing design. This device presents advanced technology with CMOS sensors and multi-spectral light sources to perform diagnostic assays, including immunochromatography, immunofluorescence, colorimetry, and dry chemistry tests. Both devices show minimal requirements for function, such as Wi-Fi or internet connection for patient registration and electrical energy for battery charging. Additionally, Hilab Flow and Lens are factory-calibrated and present individual calibration capsules to guarantee the correct functioning of sensors and focus mechanisms over time [14].

### 2.3. Sample Preparation Process

The Hilab System presents two single-use diagnostic kits (Figure 2) for sample collection and preparation. The first enables the HB quantification (Hilab Flow) through a blood collection pipette and a chromatographic strip inserted in a plastic capsule (Figure 2; Panel A). The second diagnostic kit permits the cell counts (Hilab Lens) through a disposable hemocytometer, two diluent solutions, blood collection pipettes, and sample transfer pipettes (Figure 2; Panel B). After the patient registration in the Hilab platform, the test operator collects a drop of blood (10 μL) and deposits it on the chromatographic strip, which promotes the RBC lysis and the HB conversion into methemoglobin. The capsule is closed and insertion into the Hilab Flow concludes the first preparation step. Following this, using the second diagnostic kit, the test operator collects a drop of blood (40 μL), adds it to the diluent solution 1 (dilution factor 1:10), and follows the same procedure for the diluent solution 2 (dilution factor of 1:180). After blood homogenization, the operator individually transfers a drop (10 μL) of solutions 1 and 2 to the hemocytometer chambers and then inserts the hemocytometer into the Hilab Lens. Concerning the diluent solutions, these were developed and patented by Hilab^®^ (BR 10 2021 026289 3; BR 10 2021 026290 7) and allow blood cell observation and quantification in a liquid medium (Figure 2; Panel D). The diluent solution 1 promotes the RBC lysis and WBC dye, enabling a 4-part differential. The diluent solution 2 enables RBC and PLT quantification and morphological feature observation. Both kits contain the materials needed to perform the capillary puncture (lancet, isopropyl alcohol swab, and curative; Figure 2; Panel A and B).

### 2.4. Imaging Processing and Object Classification

Hilab Lens employs a deep learning approach to detect and classify blood cells. The device uses data augmentation techniques, such as rotation and mirroring of existing images for the RBC, PLT, and WBC subpopulations, to increase the data available for training. Since a single image can contain thousands of small cells, the Hilab Lens divides each image into a grid and processes each sub-image individually using the YOLO (you only look once) deep learning approach. The YOLO model is an artificial neural network designed to be an object detector and classifier. It runs a single convolutional network on the image and thresholds the detections by the model’s confidence [15]. The Hilab dataset consists of figures labeled by our specialists, acquired from the Hilab Lens during the preliminary stages of the deployment. This dataset is updated periodically with new images from the exams performed, the model is retrained periodically, and its performance evaluation consists of F1 score, recall, and precision. Additionally, the device incorporates an overlap verification step, which considers detections with less than a threshold of intersection to be the same cell.

The Hilab Flow leverages artificial intelligence and computer vision techniques to detect hemoglobin (HB) levels on the chromatographic strip and estimate the RBC indices. Upon collecting the signals from the blood sample, the device utilizes signal processing methods to extract the mean colorimetric value of the biological sample, which is then subjected to regression analysis to determine the HB concentration. Subsequently, based on the RBC and HB outcomes, the Hilab System estimates the hematocrit (HCT), mean corpuscular volume (MCV), and mean corpuscular hemoglobin (MCH).

### 2.5. Data Processing

After the sample processing by AI, Hilab Flow and Lens use the internet to transmit the processed sample data of the patient to the Hilab Software. This software is an interactive platform developed and patented (BR 10 2017 008549 0) by Hilab, used to link qualified healthcare professionals to the AI-processed blood sample. Before enabling human analysis, the software bases the cell counting in the YOLO model, through the sample image analysis and object identification within it, along with their locations and classifications. The standard dilution factor (179-fold for RBC and PLA; 9-fold for WBC) and the high-precision hemocytometer chambers allow high-accuracy results. For Hilab Flow analysis, the software uses computer vision methods and regression analysis to calculate the HB concentration. Following this, from the HB and RBC obtained values, the Hilab Software realizes the HT, MCV, MCH, and MCHC estimative [16].

Through an exclusive access key, the health professionals connect to the Hilab Software and evaluate the AI results of each patient. In the event of discrepancies between the AI and human conclusions, the results provided by the qualified professional always supersede those produced by the AI. If the corrected results differ significantly from the initial output of the YOLO model, the differences can be used as training data to improve the model’s accuracy. The continually refining process and updating the model based on new data is known as “retraining” (Appendix A) and is considered an essential aspect of many machine learning applications. At the end of the clinical analysis, the health professional confirms the test information, and the patient or physician receives the CBC report through email or SMS. All Hilab System processes, from the sample collection to the CBC report, take 30 to 40 min. It is important to emphasize that the Hilab Software keeps the traceability of all processes, allowing access to the exam lot, test realization device, test operator register, and patient exam history. Finally, Hilab offers a 24-7 service to ensure the correct system function, report delivery, and customer support and employs two non-relational databases to prevent any correlation of information in the event of a system breach.

### 2.6. Clinical Protocol

Health-qualified professionals collected venous whole blood samples (n = 550 patients) from patients between 18 and 86 years old, including males (45%) and females (55%). The blood samples encompassed healthy (51%) and pathological conditions (49%), such as active or remission state cancer (20.43%), anemia (17.27%), cardiological diseases (7.3%), diabetes (1.5%), autoimmune diseases (1%), renal diseases (0.6%), and obesity (0.6%). Blood samples were collected, stored in sterile and single-use blood collection tubes containing K3EDTA (Vacuette^®^, Greiner Bio-One, Sao Paulo, Brazil), and processed up to 12 h after collection. After the venous whole blood collection, a blood smear of each sample was prepared and stained with May–Grunwald–Giemsa dye (Laborclin; ANVISA registration no. 10097010105), following the standard protocol [17]. The RBC morphology alterations, PLT abnormalities, and the presence of immature cells were evaluated in the conventional blood smears by trained hematologists. Finally, the accuracy, specificity, sensitivity, kappa coefficient, and balanced accuracy rate between the Hilab System and standard methodology were calculated and displayed. As this was a double-blind study, different hematologists evaluated the Hilab System, blood smear, and Sysmex XE-2100 data. A fourth professional compiled the methodologies data and carried out the statistical analysis.

### 2.7. Method Comparison

The sample processing encompassed the blood analysis in the Hilab System and the conventional hematology analyzer (XE-2100, Sysmex Corporation, Japan) for reference values establishment. For statistical analysis, the study calculated and displayed the Passing–Bablok regression, Student *t*-test, bias, and the Bland–Altman plot of each CBC analyte. In addition, the study evaluated the Hilab System accuracy compared to Sysmex XE-2100 data across the clinical range. Thus, for each analyte, the data provided by each CBC methodology were transformed into the standard reference interval and compared. Following this, through a confusion matrix, the parameters of specificity, sensitivity, balanced accuracy, and kappa coefficient were evaluated. As this was a double-blind study, different professionals employed the Hilab device and the conventional analyzer analysis. Additionally, all biological samples collected were single-use for this study and discarded after the processing, following the potentially infected samples’ standard procedure.

### 2.8. Influence of Anticoagulant and Equivalence between Venous and Capillary Blood Samples

To assess equivalence between different origins of blood, health-qualified professionals concomitantly collected capillary and venous blood samples from 150 patients between 21 and 78 years old, including males (41%) and females (59%). Due to the critical clinical condition of patients from the hematology and BMT (bone marrow transplant) intensive care unit, we invited only patients from the blood bank and emergency room to participate in this analysis. The blood samples encompassed healthy (60%) and pathological conditions (40%), such as active or remission state cancer (17%), anemia (10%), cardiological diseases (10%), diabetes (%), and obesity (1.6%). Blood samples were collected, stored in sterile and single-use blood collection tubes containing K3EDTA (Vacuette^®^, Greiner Bio-One, Brazil), processing up to 12 h after collection, and analyzed on the Hilab System device. The results obtained with the capillary blood were compared with their respective venous sample (plus K3EDTA anticoagulant) using the Bland–Altman plot and the paired Student *t*-test.

### 2.9. Flagging Study

After the venous whole blood collection, a blood smear of each sample was prepared and stained with May–Grunwald–Giemsa dye (Laborclin; ANVISA registration no. 10097010105), following the standard protocol [17]. The RBC morphology alterations, PLT abnormalities, and the presence of immature cells were evaluated in the conventional blood smears by trained hematologists. Finally, the study analyzed the accuracy, specificity, sensitivity, kappa coefficient, and balanced accuracy rate, comparing the Hilab System and standard methodology. As this was a double-blind study, the analyses of the Hilab device and stained blood smear were performed by different professionals. At the end of the study, a third professional compiled the Hilab Lens and conventional blood smear figures.

### 2.10. Statistical Analysis

All tests were double-blinded and analyzed using R software (version 4.2.1) for statistics package analysis. The Shapiro–Wilk normality test was applied to ensure the data met the criteria for parametric tests. Once accepted, we presented the data as tables, Bland–Altman or Passing–Bablok plots. The statistical analysis encompassed the average (μ), standard error (SE), bias, Pearson correlation, and Student *t*-test (paired or unpaired). *p* ≤ 0.05 was the significance level set, and no *p*-value adjustment for multiple tests was necessary. The groups (Hilab, Sysmex XE-2100, blood smear) were the independent factors used. When appropriate, we evaluated the accuracy between methodologies through a confusion matrix. In these cases, we assessed each value as inside (1) or outside (0) of the reference range [18] and calculated the specificity, sensitivity, kappa coefficient, and balanced accuracy of these methodologies. The study used the Grubbs’ test or extreme studentized deviate (ESD) method to determine the significant outliers (*p* ≤ 0.05).

### 2.11. Ethical Considerations

The study was previously approved by the Research Ethics Committee of the Paranaense League Against Cancer (CAAE number 49961421.3.1001.0098). Additionally, all methods were performed following the relevant guidelines and regulations [19].

## 3. Results

We started our analyses by comparing the Hilab System and Sysmex XE-2100 (reference methodology) data for all CBC analytes (Figure 3). Subsequently, we evaluated the accuracy between both CBC methodologies (Table 1). To evaluate the effect of anticoagulant (K3EDTA) and the equivalence between venous (conventional blood collection method) and capillary (Hilab System collection method) blood samples, we collected venous and fingerstick blood samples from 150 patients and compared their results on the Hilab System device (Figure 4). Finally, we evaluated the Hilab System flagging capability, comparing the cell alterations released by the handheld device and the standard blood smear microscopic analysis (manual microscopy; double-blinded analysis; gold-standard methodology; Figure 5; Table 2).

### 3.1. Method Comparison

The method comparison study encompassed a wide range of values for all CBC analytes (Appendix A; Appendix A), comprising data inside and outside their reference range. For Pearson correlation analysis, most analytes showed a high correlation (r > 0.8) between the Hilab System and the Sysmex XE-2100 analyzer (r values; RBC-0.94; HB-0.95; HT-0.94; MCV-0.70; MCH-0.71; MCHC-0.71; PLT-0.98; WBC-0.99; NEU-0.99; LIN-0.95; MON-0.94; EOS/BAS-0.81; Figure 3). Furthermore, both diagnostic methodologies were not statistically different (*p* > 0.05) from each other, presenting a high agreement for all analytes, as shown by the Bland–Altmann analysis (Appendix A).

Table 1 compares the Hilab System and Sysmex XE-2100 results through the accuracy, specificity, sensitivity, kappa coefficient, and balanced accuracy evaluation of all CBC analytes. Thus, for each methodology, we individually classify the data according to the standard reference interval and construct a confusion matrix for each analyte, considering 1 or 0 for values inside or outside the reference range, respectively. As a result, all analytes presented high (>89%) accuracy, specificity, sensitivity, and balanced accuracy rates. Additionally, all kappa coefficients were higher than 0.87 (RBC—0.88; HB—0.95; HT—0.95; MCV—0.87; MCH—0.92; MCHC—0.83; PLT—0.93; WBC—0.96; NEU—0.96; LIN—0.94; MON—0.93; EOS/BAS—0.93).

### 3.2. Influence of Anticoagulant and Equivalence between Venous and Capillary Blood Samples

Figure 4 shows the evaluation of anticoagulant and sample type influence on the Hilab System through compared data of venous and their respective fresh capillary blood CBC results. The study encompassed a wide range of values for all analytes (Appendix A; Appendix A), comprising data inside and outside their reference range. For all analytes, venous samples (plus K3EDTA) presented no statistical differences (*p* > 0.05) to fresh fingerstick blood samples, evaluating the paired Student *t*-test (*p* values: RBC—0.94; HB—0.79; HT—0.78; MCV—0.96; MCH—0.72; MCHC—0.42; PLT—0.40; WBC—0.14; NEU—0.53; LIN—0.22; MON—0.53; EOS/BAS—0.13). Moreover, the different sample types presented a high agreement for all analytes and low BIAS, as shown by the Bland–Altmann analysis.

### 3.3. Flagging Study

Among 550 venous samples evaluated, the study analyzed 50 positive samples for RBC morphological abnormalities. The analysis encompassed the presence of anisocytosis, microcytosis, acanthocytes, dacrocytes, and elliptocytes. Additionally, 10 positive samples for large platelets and 20 positive samples for immature cells were analyzed, covering the presence of band neutrophils and different blast cell types. Figure 5 illustrates the observation of immature cells and RBC/PLT morphological alterations in an optical microscope through a blood smear stained with May–Grunwald–Giemsa dye (reference methodology) compared to a hemocytometer with the Hilab Lens solutions. As a result, all CBC alterations presented high similarity between the Hilab Lens liquid medium and the traditional blood smear analysis.

Table 2 compares the Hilab System and standard blood smear methodology, evaluating the accuracy, specificity, sensitivity, kappa coefficient, and balanced accuracy for common blood cell alterations. Thus, we classify the observed data in both methodologies and construct a confusion matrix for all parameters, considering 1 or 0 for the presence or absence of morphological abnormalities and immature cells. As a result, all parameters presented high (>97%) accuracy, specificity, sensitivity, and balanced accuracy rates. Additionally, all kappa coefficients were higher than 0.94 (anisocytosis—0.96; microcytosis—1; acanthocytes—1; dacryocytes—1; elliptocytes—0.93; large platelets—1; immature cells—0.94).

## 4. Discussion

The CBC test is part of the everyday hospital routine, providing clinical findings for the patient diagnosis and treatment [2,20]. Based on parameters such as comparability and flagging capacity, this study provided an extensive Hilab System clinical validation in the Erasto Gaertner Hospital, a reference institution for oncological patient diagnosis and treatment.

The comparability study encompassed a substantial range of data for all analytes, evaluating healthy and diseased patients. Cancer patients in active or remission state represented the condition most analyzed in the pathological group of the study, encompassing lung, leukemia, breast, colorectal, prostate, skin, and stomach cancer cases. Patients with anemia, diabetes, obesity, cardiological, autoimmune, and renal diseases represented the other conditions of the pathological group. These data demonstrated that although the oncological patient was the focus of the study, other health conditions that affect the CBC test were evaluated, and none proved to be a limiting factor for the Hilab CBC test. Additionally, even encompassing extreme values for all CBC analytes, the Hilab device presented no significant differences (*p* > 0.05) compared to the conventional hematological analyzer (Sysmex XE-2100) data. In parallel, the high Pearson correlation values (≥0.9) found for most analytes emphasize the reliability of the Hilab System (Figure 3). Additionally, comparing the clinical range of all CBC analytes, the Hilab System and Sysmex XE-2100 presented high accuracy, sensitivity, specificity, and balanced accuracy values, as well as the kappa coefficient (≥0.95 for most analytes; Table 1).

Estimated parameters such as MCV, MCH, and MCHC presented a lower Pearson correlation (≥0.7). As the MCV, MCH, and MCHC estimation is accomplished through calculations taking RBC and HB values into account, slight differences between methodologies for their values may be subject to some discrepancies in these analytes. However, it is necessary to highlight that MCV, MCH, and MCHC presented satisfactory values for accuracy, sensitivity, specificity, and balanced accuracy (Table 1), showing that the Hilab System and Sysmex XE provide similar clinical range data and, consequently, would lead to a similar report interpretation and conduct by physicians.

Regarding the equivalence between venous (plus K3EDTA) and capillary blood samples, as previously observed [14,21,22], no analyte presented statistically significant differences (*p* > 0.05; paired *t*-test; Figure 4) from each other, proving the usefulness of the Hilab System sample collection methodology. This less-painful capillary blood collection is even more necessary for the oncologic patient, who usually presents difficult venous access, a higher rate of microbial contamination, and a constant negative experience during blood collection [23]. Thus, these data demonstrate how this handheld device can improve the patient experience for the CBC test procedure, making for a more comfortable hospital period.

The Hilab System flagging study focused on the most common CBC test alterations [24]: RBC morphological variation, large platelets, and the presence of immature cells. For all alterations evaluated, the Hilab System presented a high accuracy (>0.98 for most parameters; Table 2) compared to the conventional blood smear observation, established as a gold-standard methodology for cell analysis [25]. Additionally, we emphasize the high similarity between the cells observed in the blood smear and the liquid medium containing the Hilab staining solutions (Figure 5). These staining solutions stain specific cell structures without promoting morphological alterations, differing from commercial diluent solutions such as Hayem, Rees-Ecker, or Turk solution [26,27,28]. In this manner, the Hilab solutions maintain the blood dilution in a constant ratio, in addition to cell quantification and classification. The cell classification is similar to blood smear findings but requires fewer steps for sample preparation and is less susceptible to technical artifacts, being more simplified and user-friendly. Blood smear technique artifacts commonly occur in centralized laboratories due to health professionals’ inexperience, incorrect time of sample in EDTA tubes, and incorrect fixing or staining techniques [29,30], so the Hilab solutions may represent a considerable advantage in this field.

The intense routine of most clinical laboratories does not allow the microscopic analysis of all blood smears by a trained hematologist. With the development of automated blood-cell analyzers, the proportion of blood smear analysis in many clinical settings is below 10% and heavily dependent on “flags” produced by the device. However, the blood smear analysis remains clinically necessary for several circumstances [20,25,29,30,31]. Since this analysis is an essential tool for a differential diagnosis, the flagging study demonstrated how the Hilab System represents an advantage compared to conventional CBC devices through an AI-layer screening and specialized hematologist analysis of all blood samples.

Unlike the first Hilab System validation study [14], this work encompassed a new CBC analyte through satisfactory MCHC quantification, a relevant index for anemia diagnosis and other blood-related diseases [32]. Additionally, in this study, the evaluated CBC methodologies did not present statistical differences in EOS/BAS count. This improvement is due to the evaluation area and AI cell identification refinement implemented in the Hilab Lens. In this sense, the increased number of evaluated blood samples enhanced the system image processing and object classification resolution, leading to more concordant results between Hilab System and conventional hematology analyzers.

Although these results demonstrate the reliable clinical application of the Hilab System, evaluating a wide range of CBC values in patients from three different sectors of Erasto Gaertner Hospital, this study presents some limitations. In the second validation study of the Hilab device, even when performed with patients of a hematology-specialized hospital, we did not observe a sufficient number of such cells as blasts, metamyelocytes, and promyelocytes to allow the individual construction of a robust database for these kinds of cells. This being so, the current version of the CBC Hilab device AI can report only the presence of the band neutrophils and immature cells in general. However, since the system does not segregate the lineage or maturation rate of the cells, different immature cell types may be present in the sample when the Hilab System reports this flag and require further analyst interpretation. More testing in distinct centers is in progress to increase these cell subtypes’ databases and, consequently, the classification power of the system.

## 5. Conclusions

The presented data demonstrate considerable advantages of the Hilab System, as a fast CBC report is considered a key benefit for patients who need urgent decisions about further diagnostic and therapeutic procedures. For the hospital, the Hilab POC device does not require the sample transport process to the clinical laboratory, besides improving the patient experience through humanized and less painful blood collection. The requirement of minimally invasive POC testing is even more evident for intensive care unit (ICU) patients, in which frequent blood monitoring requires an expressive amount of samples and repeated blood-collection procedures, commonly leading them to develop anemia due to chronic clinical investigation [33]. Additionally, it is relevant to emphasize that oncological patients undergoing chemotherapy need minimal values in specific CBC analytes to realize the procedure. Thus, routine pre-chemotherapy blood tests should be undertaken 2–3 days before the treatment to avoid unwarranted delays [34]. Therefore, the fast and less painful Hilab CBC test may represent an essential advantage for these patients and the hospital routine. Based on the elevated costs associated with current available automated analyzer usage, which involves trained laboratory technicians, laboratory structure, and commercial hematological controls, the results provided by the Hilab device present relevant advantages for the hematological POCT field.

## Figures and Tables

**Figure 1 diagnostics-13-01695-f001:**
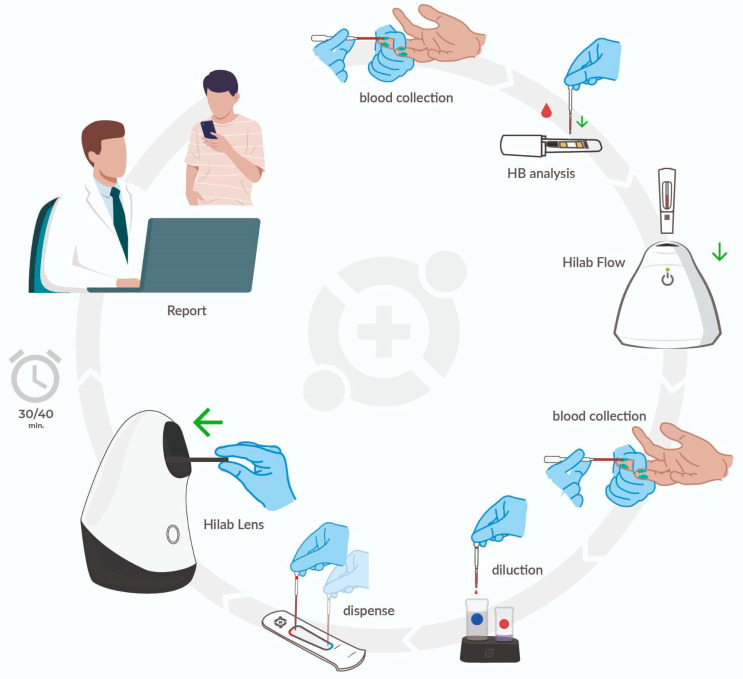
The Hilab System POCT hematology analyzer. The test operator performs the patient capillary puncture with a lancet. Following this, the operator collects a drop of blood (10 μL) through a disposable pipette and deposits it on a chromatographic strip contained in a plastic capsule. The capsule insertion into the Hilab Flow (chromatography strips analyzer) device concludes the HB quantification process. Next, the operator collects the second drop of the sample (80 μL) and deposits it in the Hilab Lens diluent solutions for blood staining and dilution. Finally, the test operator transfers a drop of each solution into the chambers of the disposable hemocytometer and inserts it into the Hilab Lens device (handheld microscopy). Two (Hilab Flow; Hilab Lens) single-use test kits provide the materials used for CBC Hilab test processing, which accompany isopropyl alcohol swabs and curative. In approximately 30 min, the patient and/or physician receive the report signed by Hilab technical responsible and a laboratory analyst through email or SMS.

**Figure 2 diagnostics-13-01695-f002:**
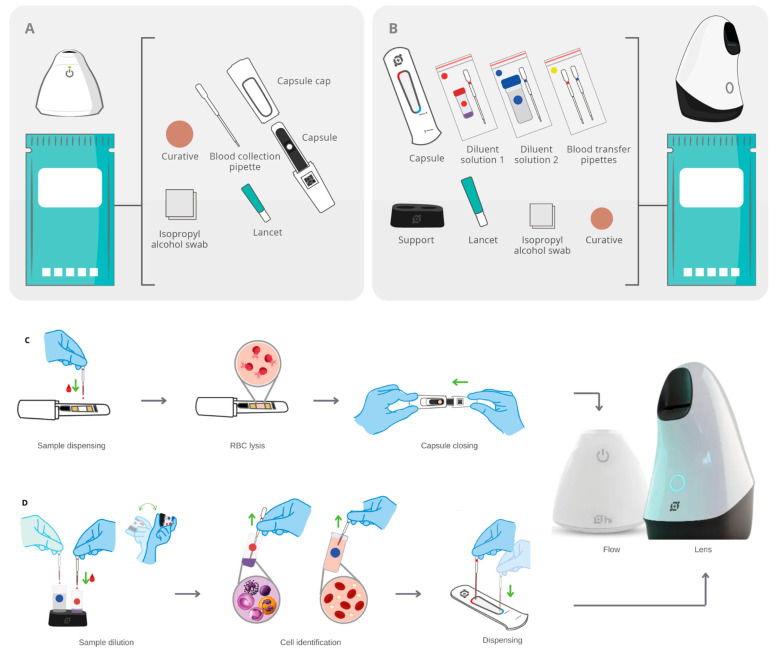
The single-use diagnostic kits of the Hilab System and sample preparation process. (**A**) Components of the Hilab Flow test kits: capsule, capsule cap, blood collection pipette, lancet, isopropyl alcohol swab, and curative. (**B**) Components of the Hilab Lens test kits: capsule (disposable hemocytometer), blood collection pipettes and blood transfer pipettes, mixing bottle of diluent 1 (WBC count), mixing bottle of diluent 2 (RBC and PLT count), support for mixing bottles, lancet, isopropyl alcohol swab and curative. (**C**) The test operator collects a drop of blood (10 μL) and deposits it on the chromatographic strip, which promotes the RBC lysis and the HB conversion into methemoglobin. The capsule closing and insertion on the Hilab Flow concludes the process. (**D**) The test operator collects a drop of blood (40 μL) and adds it to the diluent solution 1 (red circle; dilution factor 1:10), which promotes the RBC lysis and differential WBC dye. The diluent solution 2 (blue circle; dilution factor of 1:180) follows the same procedure, enabling the RBC and PLT visualization. After blood homogenization, the operator individually transfers a drop (10 μL) of solutions 1 and 2 to the hemocytometer chambers and then inserts the hemocytometer into the Hilab Lens.

**Figure 3 diagnostics-13-01695-f003:**
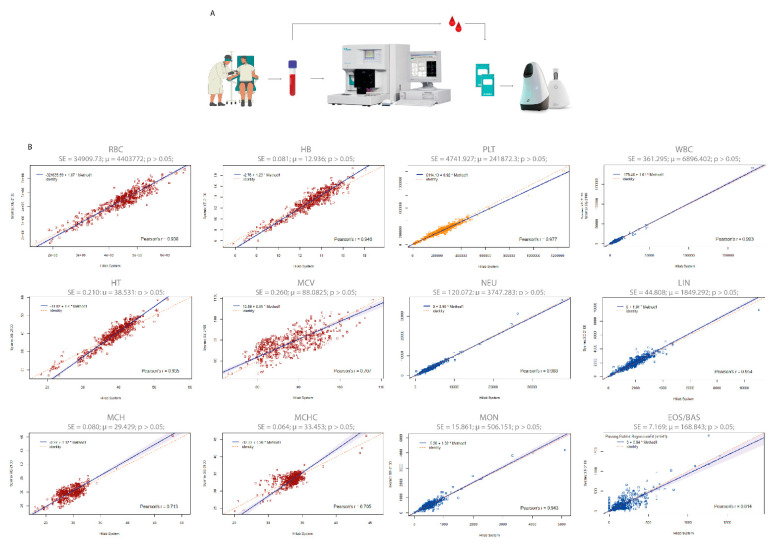
Passing–Bablok regression plot of the method comparison study between the Hilab System and the Sysmex XE-2100. (**A**) Clinical protocol for method comparison analysis. (**B**) Average (μ), standard error (SE), and Student-*t* test *p*-values are demonstrated for each CBC analyte (n = 550 patients/group). Red graphs represent the analytes related to red blood cells (RBC, HB, HT, MCV, MCH, MCHC), yellow graphs to platelets (PLT), and blue graphs to white blood cells (WBC, NEU, LIN, MON, EOS/BAS).

**Figure 4 diagnostics-13-01695-f004:**
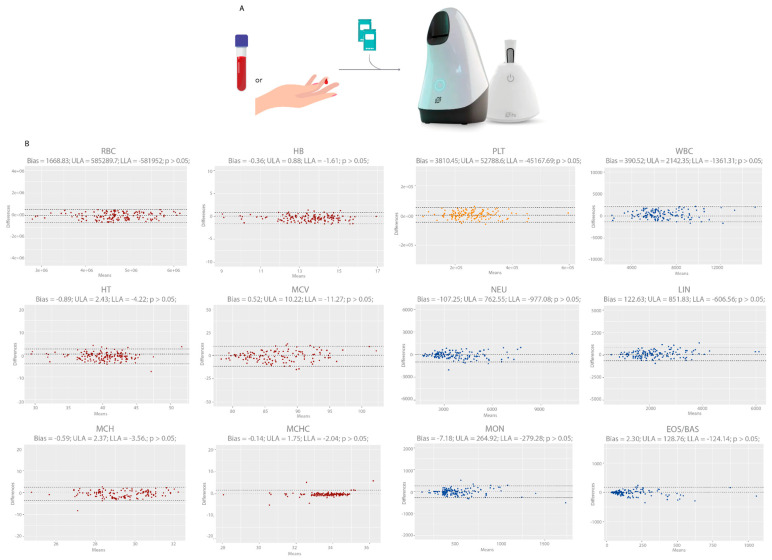
Bland–Altman plot of Hilab System results for venous (plus K3EDTA) versus fingerstick blood samples. (**A**) Clinical protocol for anticoagulant influence analysis and equivalence between venous and capillary blood samples. (**B**) Graphs demonstrate bias, Student paired *t*-test *p*-value, the upper limit of agreement (ULA), and lower limit of agreement (LLA) for each CBC analyte (n = 150 patients/group). Red graphs represent the analytes related to red blood cells (RBC, HB, HT, MCV, MCH, MCHC), yellow graphs to platelets (PLT), and blue graphs to white blood cells (WBC, NEU, LIN, MON, EOS/BAS).

**Figure 5 diagnostics-13-01695-f005:**
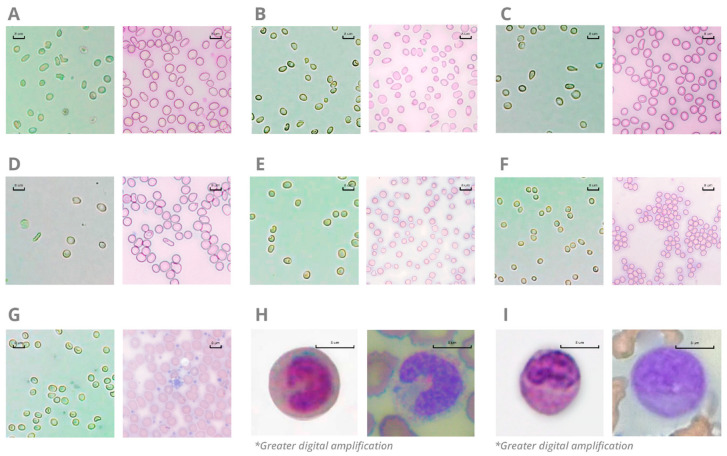
Comparison between Hilab System (left) and standard blood smear methodology (right) for blood cell alterations. Figures show the presence of acanthocytes (Panel (**A**)), anisocytosis (Panel (**B**)), dacrocytes (Panel (**C**)), elliptocytes (Panel (**D**)), macrocytosis (Panel (**E**)), microcytosis (Panel (**F**)), large platelets (Panel (**G**)), band neutrophils (Panel (**H**); greater digital amplification for better nucleus visualization), and immature cells (Panel (**I**); greater digital amplification for better nucleus visualization) in blood samples. Blood smears were stained with May–Grunwald–Giemsa dye, and the images were captured by a Global Optics (NO115T1, Global Trade Technology, São Paulo, Brazil) optical microscope (original magnification ×1000).* Images with greater digital amplification.

**Table 1 diagnostics-13-01695-t001:** Hilab System versus Sysmex XE-2100 accuracy, specificity, sensitivity, kappa coefficient, and balanced accuracy for all CBC analytes.

Analyte	Accuracy (%)	Specificity (%)	Sensitivity (%)	Kappa coefficient	Balanced Accuracy (%)
RBC	97.0	98.0	89.7	0.88	92.7
HB	98.8	99.7	94.0	0.95	96.9
HT	98.1	97.9	98.3	0.95	98.1
MCV	96.9	97.2	95.3	0.87	96.2
MCH	98.1	98.0	98.7	0.92	98.3
MCHC	95.5	99.1	83.3	0.83	91.3
PLT	97.5	97.3	98.2	0.93	97.8
LEU	98.9	99.3	96.7	0.96	98.0
NEU	99.1	99.2	98.5	0.96	98.8
MON	98.9	99.9	99.9	0.93	94.5
LIN	98.7	99.3	95.1	0.94	97.2
EOS/BAS	98.0	99.0	89.0	0.93	94.5

**Table 2 diagnostics-13-01695-t002:** Hilab System versus peripheral blood smear stained with May–Grunwald–Giemsa: Accuracy, specificity, sensitivity, kappa coefficient, and balanced accuracy.

Analyte	Accuracy (%)	Specificity (%)	Sensitivity (%)	Kappa Coefficient	Balanced Accuracy (%)
Anisocytosis	99.4	99.8	96.0	0.96	97.9
Microcytosis	100.0	100.0	100.0	1	100.0
Acanthocytes	100.0	100.0	100.0	1	100.0
Dacryocytes	100.0	100.0	100.0	1	100.0
Elliptocytes	99.5	99.6	95.4	0.93	97.5
Large platelets	100.0	100.0	100.0	1	100.0
Immature cells	99.6	99.6	100.0	0.94	99.8

## Data Availability

All data related to this paper may be requested from the author alexia.gasparin@hilab.com.br.

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
