# Peer review of "Hilab System Device in an Oncological Hospital: A New Clinical Approach for Point of Care CBC Test, Supported by the Internet of Things and Machine Learning"

_diagnostics, 2023, doi:10.3390/diagnostics13101695_

Round 1

Reviewer 1 Report

Dear authors,

I have gone through your paper. The idea is interesting and novel. However, following changes must be incorporated for the paper to ready for acceptance. The paper must be strengthened from machine learning view point.

Abstract: Good. However, please what machine learning and IoT is in the abstract.

Introduction: Machine learning and IoT details are missing again. Add a paragraph or two about these technologies before you introduce Hilab (Third paragraph). Further, literature review is missing . CBC markers and ML have been used in a lot of medical applications for diagnosis and prognosis. It is recommended to review the following literature since all the applications use ML and CBC markers 

1. Chadaga K, Prabhu S, Bhat V, Sampathila N, Umakanth S, Chadaga R. A Decision Support System for Diagnosis of COVID-19 from Non-COVID-19 Influenza-like Illness Using Explainable Artificial Intelligence. Bioengineering. 2023 Mar 31;10(4):439.

2. Khanna VV, Chadaga K, Sampathila N, Prabhu S, Bhandage V, Hegde GK. A Distinctive Explainable Machine Learning Framework for Detection of Polycystic Ovary Syndrome. Applied System Innovation. 2023 Feb 23;6(2):32.

3. Chadaga K, Prabhu S, Sampathila N, Chadaga R, KS S, Sengupta S. Predicting cervical cancer biopsy results using demographic and epidemiological parameters: a custom stacked ensemble machine learning approach. Cogent Engineering. 2022 Dec 31;9(1):2143040.

You can include other diseases and CBC/ML too.. Literature review must be made strong.

Figure 1 is good!

Mention the contributions and novelty point wise in the introduction section. Also, mention some research gaps prior to this.

2. Materials and Methods.

2.1 .. Mention some more things about the dataset . Add an image or table to describe the parameters if possible

2.2 Good

2.3 Good

2.4-2.9 Good

2.10- Mention some statistical tables to give some idea descriptive statistics if possible.

2.11 Excellent since you are working on a novel data

3. Results

The tables and figures must be explained in a better way. Add atleast three to four sentences about the figures and tables. The figures are execllent!

3.1 to 3.3 - Okay , good!

4. Discussion -Okay

Add a conclusion section, it is necessary. Also mention limitations and future work details. This is necessary.

Overall verdict: The manuscript is written well, offers novelty and is truly useful for the medical and data science community. Please make the above changes and resubmit.

Author Response

Reviewer 1

Abstract: Good. However, please what machine learning and IoT is in the abstract.

First, we thank you for your kind consideration. In the new version of the manuscript, we reviewed this point. Please, see page 1, section “Abstract”.

Introduction: Machine learning and IoT details are missing again. Add a paragraph or two about these technologies before you introduce Hilab (Third paragraph). Further, literature review is missing . CBC markers and ML have been used in a lot of medical applications for diagnosis and prognosis. It is recommended to review the following literature since all the applications use ML and CBC markers. 

We appreciate the constructive observations and questions raised. In the new version of the manuscript, we reviewed this point. Please, see page 2, section “Introduction”, third paragraph. 

Figure 1 is good! Mention the contributions and novelty point wise in the introduction section. Also, mention some research gaps prior to this.

Once again, we thank the constructive observations. In the new version of the manuscript, we reviewed this point. Please, see page 2, section “Introduction”, third and fourth paragraph. 

  1. Materials and Methods: 2.1 .. Mention some more things about the dataset . Add an image or table to describe the parameters if possible

We added some descriptive statistics as supplemental material, in tables S1 and S2. Please see Supplementary Materials, Table S1 - Descriptive statistics for method comparison study; and Table S2 - Descriptive statistics for the equivalence study between venous and capillary blood samples).

2.10- Mention some statistical tables to give some idea of descriptive statistics if possible.

We added some descriptive statistics parameters as tables S1 and S2. Please see Supplementary Materials, Table S1 - Descriptive statistics for method comparison study; and Table S2 - Descriptive statistics for the equivalence study between venous and capillary blood samples).

  1. Results. The tables and figures must be explained in a better way. Add atleast three to four sentences about the figures and tables. The figures are execllent!

Once again, we thank the constructive observations. In the new version of the manuscript, we reviewed this point. Please, see section Results.

Add a conclusion section, it is necessary. Also mention limitations and future work details. This is necessary.

As suggested, a conclusion section was added in the new version of the manuscript. Also, we mentioned limitations and future work details. Please, see the last paragraph of the discussion topic, besides the conclusion section.

Reviewer 2 Report

This paper discusses the Hilab System device, which is a new approach for point-of-care CBC testing in oncological hospitals. The device is supported by the Internet of Things (IoT) and machine learning technologies, which enable real-time monitoring and analysis of patient data. The paper explores the advantages of the Hilab System device over traditional CBC testing methods, such as improved accuracy, faster results, and greater convenience for patients.

Abstract

Overall, the abstract provides a clear and concise summary of the study's objectives, methods, and main findings.

Introduction

The introduction provides a clear overview of the complete blood count (CBC) test and its importance in clinical practice. It highlights the limitations of conventional hematological analyzers, including high initial costs, structured laboratory environments, and subsequent microscopic evaluations. The introduction then introduces the Hilab System, a novel point-of-care hematological platform that uses microscopy and chromatography techniques, combined with machine learning and artificial intelligence. The study evaluated the accuracy and flagging capabilities of the Hilab System compared to the CBC results provided by Sysmex XE-2100 in a reference institution for the diagnosis and treatment of oncological patients. The introduction is well-organized and provides sufficient context for the study. However, It is suggested that you cite the following works related to CBC and COVID-19:

https://doi.org/10.1016/j.dib.2023.109016

https://doi.org/10.1016/j.meegid.2022.105228.

Material and methods

The Materials and Methods section describes the experimental design, informed consent, the Hilab System device, sample preparation process, imaging processing, and object classification. It is necessary to include a graph as a pipeline focusing on the application of IoT and Machine Learning

Results and Discussion

The results are presented in a clear and concise way, clearly evidencing the findings. The discussion is well written and includes several interesting comparisons. Some parts of figure 3B are not displayed correctly.

Conclusion

The conclusion section is not available

Other

The design of the figures used in the manuscript is of a high quality. Congratulations.

Author Response

Reviewer 2

After comparing in the database, we found that the following sections have highly similarity rate with your previous work (https://doi.org/10.1038/s41598-022-13913-8). and we would kindly request rephrasing during the revision:

- 2.2. The Hilab System device (line 100-118)

We appreciate the suggestion. In the new version of the manuscript, please see the section “The Hilab System device”.

- 2.3 Sample preparation process (line 123-129)

We appreciate the suggestion. In the new version of the manuscript, please see the section “The Hilab System device”.

- Figure 2 caption

Sorry for this mistake. We adjusted the figure. Please see page 5, Figure 2, “The single-use diagnostic kits of the Hilab System and sample preparation process”.

- 2.5. Data processing (line 187-19)

Once again, we thank the constructive observations. In the new version of the manuscript, please see the section “Data processing”.

The introduction provides a clear overview of the complete blood count (CBC) test and its importance in clinical practice. It highlights the limitations of conventional hematological analyzers, including high initial costs, structured laboratory environments, and subsequent microscopic evaluations. The introduction then introduces the Hilab System, a novel point-of-care hematological platform that uses microscopy and chromatography techniques, combined with machine learning and artificial intelligence. The study evaluated the accuracy and flagging capabilities of the Hilab System compared to the CBC results provided by Sysmex XE-2100 in a reference institution for the diagnosis and treatment of oncological patients. The introduction is well-organized and provides sufficient context for the study. However, It is suggested that you cite the following works related to CBC and COVID-19:

https://doi.org/10.1016/j.dib.2023.109016 and https://doi.org/10.1016/j.meegid.2022.105228.

We thank the constructive observations and questions raised. We inserted the suggested information in the “Introduction” section. Please, see page 2, third paragraph. 

The Materials and Methods section describes the experimental design, informed consent, the Hilab System device, sample preparation process, imaging processing, and object classification. It is necessary to include a graph as a pipeline focusing on the application of IoT and Machine Learning

Once again, we appreciate the suggestion. We inserted the suggested information in the “Material and methods” section, topic “data processing”. Besides, a graphical illustration of the ML process was inserted in the supplementary data. Please see supplementary data 4.

The results are presented in a clear and concise way, clearly evidencing the findings. The discussion is well written and includes several interesting comparisons. Some parts of figure 3B are not displayed correctly.

Sorry for this mistake. In the new version of the manuscript we adjusted these parameters. Please, see page 9.

The conclusion section is not available

Once again, sorry for this mistake. In the new version of the manuscript we added this topic.. Please, see page 17, conclusion section.

Reviewer 3 Report

Gasparin's manuscript utilized the Hilab System to conduct CBC tests on clinical samples, which included significant hematological changes. The authors found that the Hilab System's performance was comparable to traditional methods. However, the results suggested that the system may exhibit bias towards certain cell types/clinical conditions. Additionally, a major concern arose as the data did not appear to follow a normal distribution. This presents a challenge as many statistical methods, including Pearson correlation and Student's t-test, assume normal distribution.

Author Response

Reviewer 3

Gasparin's manuscript utilized the Hilab System to conduct CBC tests on clinical samples, which included significant hematological changes. The authors found that the Hilab System's performance was comparable to traditional methods. However, the results suggested that the system may exhibit bias towards certain cell types/clinical conditions. Additionally, a major concern arose as the data did not appear to follow a normal distribution. This presents a challenge as many statistical methods, including Pearson correlation and Student's t-test, assume normal distribution.

We understand this question and thank the constructive observations and questions raised. The Shapiro-Wilk normality test was applied to ensure that all data met the criteria for parametric tests. Once accepted, we presented the data as tables, Bland–Altman or Passing–Bablok plots. We emphasized this point in the new version of the manuscript. Please, see “Statistical analysis” section, page  8.

Reviewer 4 Report

The paper looks very well laid out in all its parts. No major issues to signal.

Just a minor detail to add in the text at line 256: "p ≤ 0.05 was the significance level set". Please specify whether p-value adjustment for multiple tests was made and which adjustment procedure was performed

Author Response

Reviewer 4

The paper looks very well laid out in all its parts. No major issues to signal. Just a minor detail to add in the text at line 256: "p ≤ 0.05 was the significance level set". Please specify whether p-value adjustment for multiple tests was made and which adjustment procedure was performed

We thank the constructive observation. As suggested, we specify the p-value adjustment in the “Statistical analysis” section. Please, see page 8.